

# Kisspeptin receptor agonist (FTM080) increased plasma concentrations of luteinizing hormone in anestrous ewes

Brian K. Whitlock[1], Joseph A. Daniel[2], Lisa L. Amelse[1], Valeria M. Tanco[3], Kelly A. Chameroy[1] and F. Neal Schrick[4]

[1] Department of Large Animal Clinical Sciences, The University of Tennessee, Knoxville, TN, United States
[2] Department of Animal Science, Berry College, Mt. Berry, GA, United States
[3] Department of Small Animal Clinical Sciences, The University of Tennessee, Knoxville, TN, United States
[4] Department of Animal Science, The University of Tennessee, Knoxville, TN, United States

Corresponding author
Brian K. Whitlock,
bwhitloc@utk.edu

## ABSTRACT

Kisspeptin receptor (KISS1R) agonists with increased half-life and similar efficacy to kisspeptin *in vitro* may provide beneficial applications in breeding management of many species. However, many of these agonists have not been tested *in vivo*. These studies were designed to test and compare the effects of a KISS1R agonist (FTM080) and kisspeptin on luteinizing hormone (LH) *in vivo*. In experiment 1 (pilot study), sheep were treated with FTM080 (500 pmol/kg BW) or sterile water (VEH) intravenosuly. Blood was collected every 15 min before (1 h) and after (1 h) treatment. In experiment 2, sheep were treated with KP-10 (human Metastin 45-54; 500 pmol/kg BW), one of three dosages of FTM080 (500 (FTM080:500), 2500 (FTM080:2500), or 5000 (FTM080:5000) pmol/kg BW), or VEH intravenously. Blood was collected every 15 min before (1 h) and after (4 h) treatment. In experiment 1, FTM080:500 increased ($P < 0.05$) plasma LH concentrations when compared to VEH. The area under the curve (AUC) of LH following FTM080:500 treatment was also increased ($P < 0.05$). In experiment 2, plasma LH concentrations increased ($P < 0.05$) following treatment with KP-10 and FTM080:5000 when compared to VEH and FTM080:500. The AUC of LH following KP-10 was greater than ($P < 0.05$) all other treatments and the AUC of LH following FTM080:5000 was greater than ($P < 0.05$) all treatments except KP-10. These data provide evidence to suggest that FTM080 stimulates the gonadotropic axis of ruminants *in vivo*. Any increased half-life and comparable efficacy of FTM080 to KP-10 *in vitro* does not appear to translate to *in vivo* in sheep.

## INTRODUCTION

Reproduction requires precise interactions among the brain, the pituitary, and the gonads (gonadotropic-axis). Gonadotropin-releasing hormone (GnRH) from the hypothalamus has long been considered a major regulator of the gonadotropic-axis because it controls

the secretion of the gonadotropins luteinizing hormone (LH) and follicle-stimulating hormone from the pituitary, the production of sexual hormones, and gametes by the gonads (*Knobil, 2005*). Until recently many unknowns remained in our understanding of the neuroendocrine mechanisms controlling the release of GnRH. For example, in female mammals estradiol induces a strong increase in GnRH release during the follicular phase of the estrous cycle, leading to preovulatory GnRH and LH surges and ultimately ovulation (*Xia et al., 1992*; *Moenter et al., 1991*; *Sarkar et al., 1976*; *Clarke et al., 1989*). However, the pathways and the hierarchy of mechanisms involved in the regulation of GnRH release are only partially understood.

Kisspeptin recently emerged as a major regulator of the gonadotropic-axis located "upstream" of the GnRH cell population in the hypothalamus. Kisspeptin was first discovered and noted for its role in the inhibition of cancer cell metastasis (*Lee et al., 1996*). The central functions of kisspeptin and the kisspeptin receptor (KISS1R) in regulating mammalian reproductive development and fertility were unnoticed until 2003, when three groups independently reported the presence of deletion and inactivating mutations of KISS1R in humans (*Seminara et al., 2003*; *De Roux et al., 2003*) and mice (*Seminara et al., 2003*; *Funes et al., 2003*) suffering from hypogonadotropic hypogonadism; a syndrome characterized by delayed or absent pubertal development secondary to gonadotropin deficiency. It was later discovered that kisspeptin was necessary for GnRH release from the hypothalamus and subsequent secretion of LH from the anterior pituitary (*Seminara et al., 2003*; *De Roux et al., 2003*; *Funes et al., 2003*; *Roseweir et al., 2009*). Kisspeptin is involved in sexual differentiation of the brain, timing of puberty, regulation of gonadotropin secretion by gonadal hormones, and the control of fertility by metabolic and environmental cues (*Roa, Navarro & Tena-Sempere, 2011*). With the recognition that kisspeptin is a major regulator of the gonadotropic-axis and a potent stimulator of gonadotropin secretion has come the notion that it could be used to manipulate reproduction. Indeed, exogenous administration of kisspeptin has been shown to stimulate LH and follicle-stimulation hormone secretion in many species (*Tena-Sempere, 2006*). However, few experiments have been designed to evaluate the potential of kisspeptin to manipulate reproduction in animals. Intravenous infusion of kisspeptin to sheep in the non-breeding season elevated gonadotropin secretion and caused ovulation (*Caraty et al., 2007*). The demonstration that intravenous infusion of kisspeptin can stimulate ovulation in seasonally anestrous female sheep offers a potential means of manipulating the reproductive axis and ultimately fertility in a multitude of species. However, kisspeptin may be of limited clinical use because of the short circulating half-life (*Kotani et al., 2001*; *Dhillo et al., 2005*; *Plant, Ramaswamy & Dipietro, 2006*; *Liu et al., 2013*). Rational modification of KISS1R agonists were synthesized to be resistant to matrix metalloproteinase (MMP) activity and/or found to have increased half-life in murine serum (FTM145 and FTM080), and to have comparable binding affinity and efficacy *in vitro* to kisspeptin (*Tomita et al., 2008*). However, *in vivo* activities of these peptides have not yet been studied. Thus, the present experiments were designed to determine and compare the effect of a novel KISS1R

agonist (FTM080) (*Tomita et al., 2008*; *Tomita et al., 2007*) and kisspeptin on plasma LH concentrations in seasonally anestrus female sheep.

## MATERIALS AND METHODS

All procedures were approved by the Berry College (Rome, GA) Institutional Animal Care and Use Committee (Protocol No. 2011/12-010). Adult parous Katahdin female sheep were used in this experiment. Sheep were housed at the Ruminant Research Unit at Berry College (Latitude = 34°18′8.33″N; Longitude = 85°11′45.29″W), exposed to average ambient temperature (25 °C average daily temperature) and summer photoperiod (14:10 (L:D) h) throughout the experiments (June), and fed a maintenance diet calculated to meet 100% of daily requirements (*National Research Council, 1985*). During the experiments sheep were kept in individual pens (1.2 × 1.2 m) to facilitate IV injection and serial blood collections.

The effects of a novel KISS1R agonist (FTM080: 4-fluorobenzoyl-Phe-Gly-Leu-Arg-Trp-NH$_2$; Graduate School of Pharmaceutical Sciences, Kyoto University) (*Tomita et al., 2008*; *Tomita et al., 2007*) and KP-10 (a biologically active C-terminally amidated cleavage fragment of kisspeptin, human Metastin 45-54, 4389-v; Peptide Institute Inc., Osaka, Japan) on plasma LH concentrations in anestrous sheep was tested. To reduce the influence of sex steroids on the kisspeptin-KISS1R system, studies were conducted during a long photoperiod to increase the likelihood of ewes being anestrous (*Smith et al., 2007*). In addition, blood samples were collected before, during, and after the experiments (7 days between samples over three consecutive weeks) and assayed to determine progesterone concentrations. Data from animals with circulating progesterone concentrations greater than 1 ng/mL (indicating active luteal tissue and therefore cyclicity) were excluded from the analysis. To facilitate treatment administration and blood sampling, each animal was fitted with an indwelling intravenous jugular catheter the day before experimentation.

Experiment 1: eight sheep (41.6 ± (SEM) 1.3 kg) were treated with FTM080 (500 pmol/kg BW; FTM080:500; diluted in sterile water; $n = 4$) or sterile water (Vehicle; VEH; $n = 4$) in a 2-mL bolus via the jugular cannula. Serial blood samples (every 15 min; 3-mL each) were collected before (for 1 h) and after (for 1 h) treatment. Blood was collected into tubes containing 7.5 mg EDTA. Plasma was stored at −20 °C for radioimmunoassay (RIA) of LH and progesterone.

Experiment 2: twenty-one sheep (48.2 ± 5.1 kg) were used in this experiment. Sheep received one of five treatments (sterile water (Vehicle; VEH; $n = 5$), KP-10 diluted in sterile water (500 pmol/kg BW; $n = 4$), or FTM080 diluted in sterile water (500 (FTM080:500; $n = 4$), 2500 (FTM080:2500; $n = 4$), or 5000 (FTM080:5000; $n = 4$) pmol/kg BW)) in 2-mL bolus via the jugular cannula. Samples were collected and handled in the same manner as described in Experiment 1 except blood samples were collected for a total of 4 h after treatment.

Plasma LH concentrations were assayed by double-antibody RIA using materials supplied by the National Hormone and Pituitary Program of NIDDK as previously described (*Coleman et al., 1993*). Limit of detection and intra-assay and inter-assay coefficient of

variance were 0.125 ng/mL and 5.5% and 9.9% for the LH assays, respectively. Plasma progesterone concentrations were determined using the Coat-a-Count® Progesterone RIA kit (Siemens, Los Angeles, CA, USA) (*Minton et al., 1991*; *Srikandakumar et al., 1986*; *Reimers et al., 1991*; *Colazo et al., 2008*). Limit of detection and intra-assay coefficients of variance for the progesterone assay were 0.1 ng/mL and 14.9%, respectively.

For Experiments One and Two circulating concentrations of LH were tested for effect of treatment, time, and treatment by time interaction using ANOVA procedures for repeated measures with JMP Software (version 7; SAS Inst. Inc., Cary, NC). Area under the LH concentration curve pre ($-60$ to 0 min) and post (0 to 60 min) treatment was calculated using the trapezoid method with MSExcel Software. Area under the LH curve was tested for effect of treatment, period (pre- or post-treatment), and treatment by period interaction using ANOVA procedures for repeated measures with JMP Software (version 7; SAS Inst. Inc., Cary, NC). Means separation was performed using Student's $T$ test when appropriate.

## RESULTS

Experiment 1: two ewes (one per treatment) were excluded from the analysis and results because their plasma progesterone concentrations were greater than 1 ng/mL (2.60 and 1.70 ng/mL) thus plasma LH concentrations from six ewes ($n = 3$) were analyzed and reported. Plasma progesterone concentrations for the remainder of the animals were less than 1 ng/mL ($0.12 \pm 0.08$ (SEM) ng/mL). Mean $\pm$ SEM plasma LH concentrations before treatment were $0.31 \pm 0.16$ ng/mL and $0.14 \pm 0.06$ ng/mL for animals treated with VEH and FTM080:500, respectively. Mean plasma LH concentrations for the period after treatment were $0.24 \pm 0.04$ ng/mL and $0.73 \pm 0.08$ ng/mL for animals treated with VEH and FTM080:500, respectively. There was an effect of time ($P = 0.0019$) and a treatment by time interaction ($P = 0.0009$) on plasma LH concentrations. Plasma LH concentrations for FTM080:500 treated animals were greater than ($P < 0.05$) VEH from 0 to 45 min following treatment (Fig. 1A).

An effect of period (pre- and post-treatment) ($P = 0.0464$) and a period by treatment interaction ($P = 0.0150$) was found when analyzing the area under the LH curve. The area under the curve of LH for FTM080:500 treated animals was greater than ($P < 0.05$) VEH from 0 to 60 min following treatment (Fig. 1B).

Experiment 2: one ewe (VEH group) was excluded from the analysis and results because of high plasma progesterone concentrations (7 days post-experiment; 2.50 ng/mL). Plasma LH concentrations from a total of 20 ewes ($n = 4$) were analyzed and reported. Plasma progesterone concentrations for the 20 animals included in the analysis were less than 1 ng/mL ($0.16 \pm 0.01$ ng/mL) before and after the experiment. Mean $\pm$ SEM plasma LH concentrations were $0.59 \pm 0.37$ ng/mL, $0.78 \pm 0.38$ ng/mL, $0.43 \pm 0.24$ ng/mL, $0.58 \pm 0.37$ ng/mL, and $0.53 \pm 0.28$ ng/mL before treatment with VEH, KP-10, FTM080:500, FTM080:2500, and FTM080:5000 pmol/kg, respectively. Mean plasma LH concentrations were $1.35 \pm 0.20$ ng/mL, $1.93 \pm 0.37$ ng/mL, $0.97 \pm 0.13$ ng/mL, $0.94 \pm 0.13$ ng/mL, and $1.29 \pm 0.29$ ng/mL for the period after treatment with VEH, KP-10, FTM080:500, FTM080:2500, and FTM080:5000 pmol/kg, respectively. There was an effect of treatment
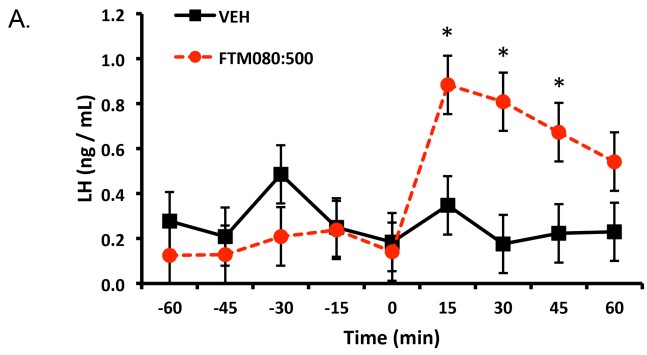

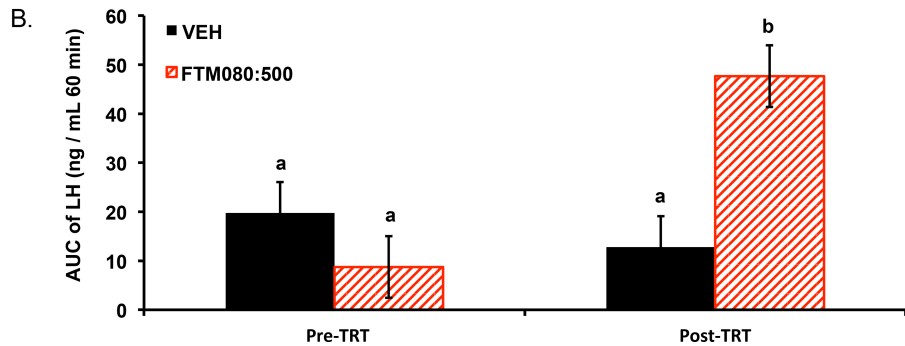

**Figure 1 Effect of i.v. KISS1R agonist, FTM080, on plasma LH concentrations in anestrous ewes ($n = 3$).** (A) Response of circulating concentration of LH (mean ± pooled SEM = 0.13) to i.v. administration of VEH and FTM080 (500 pmol/kg BW; FTM080:500). There was an effect of time ($P = 0.0019$) and an interaction for FTM080 by time for LH ($P = 0.0009$). *$p < 0.05$ vs. VEH. (B) Effect of i.v. administration of VEH and FTM080 (500 pmol/kg BW; FTM080:500) on AUC of LH concentrations from −60 to 0 min before (Pre-TRT) and from 0 to 60 min following treatment (Post-TRT) (mean ± pooled SEM = 6.29). AUCs with different superscripts differ ($p < 0.05$).

($P = 0.0134$) on mean plasma LH concentrations. Mean plasma LH concentration for the period following treatment with KP-10 (1.93 ± 0.37 ng/mL) was greater than all treatments except FTM080:5000. There was also an effect of time ($P < 0.0001$) and an interaction of treatment and time ($P < 0.0001$) on plasma LH concentrations. Plasma LH concentrations following treatment with KP-10 were greater than ($P < 0.05$) the VEH through the 45-min sample, FTM080:500 at the 30- and 45-min samples, and FTM080:2500 at 30-min (Fig. 2A). Plasma LH concentrations following FTM080:5000 was greater than ($P < 0.05$) VEH through the 30-min sample and FTM080:500 at the 15-min samples (Fig. 2A).

There was an effect of treatment ($P < 0.0001$), period (pre-treatment (−60 to 0 min); 1 h post-treatment (0 to 60 min); $P < 0.0001$), and an interaction of treatment and period ($P < 0.0001$) on area under the curve (AUC) of plasma LH concentrations. The 1 h post-treatment AUC of LH following KP-10 was greater than ($P < 0.05$) all other treatments and the 1 h post-treatment AUC of LH following FTM080:5000 was greater than ($P < 0.05$) all treatments except KP-10 (Fig. 2B). The AUC of LH in the 1 h

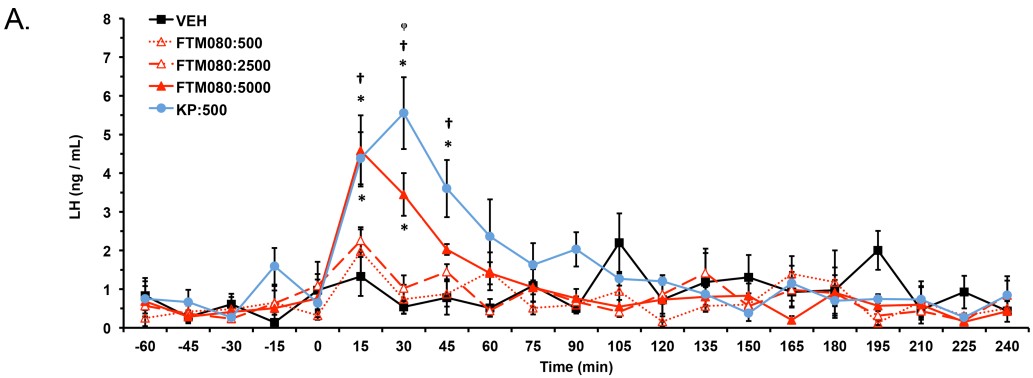

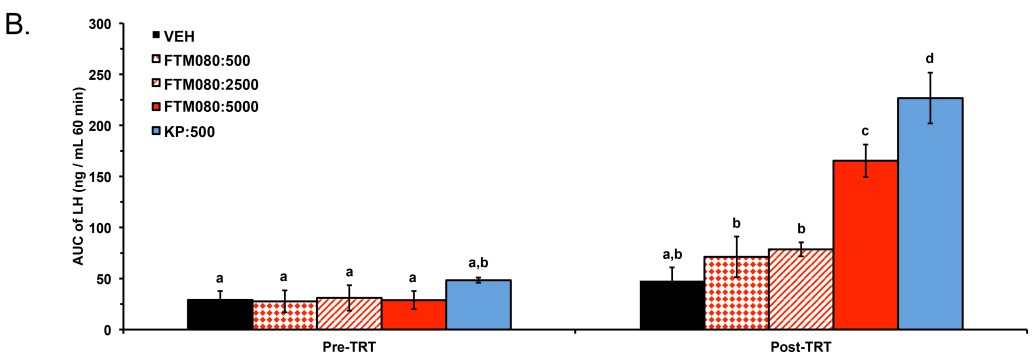

**Figure 2 Effect of i.v. KP-10 and FTM080, KISS1R agonist, on plasma LH concentrations in anestrous ewes ($n = 4$).** (A) Response of circulating concentration of LH (mean ± SEM) to i.v. administration of VEH (sterile water), KP-10 (500 pmol/kg), and FTM080 (500 (FTM080:500), 2500 (FTM080:2500), or 5000 (FTM080:5000) pmol/kg BW). There was an effect of time ($P < 0.0001$) and an interaction of treatment and time ($P < 0.0001$) on plasma LH concentrations. *$p < 0.05$ vs. VEH. †$p < 0.05$ vs. FTM080:500. $\varphi$ $p < 0.05$ vs. FTM080:2500. (B) Effect of i.v. administration of VEH (sterile water), KP-10 (500 pmol/kg BW), and FTM080:500, FTM080:2500, or FTM080:5000 on AUC of plasma LH concentrations from −60 to 0 min before (Pre-TRT) and from 0 to 60 min following treatment (1 h Post-TRT) (mean ± SEM). There was an effect of treatment ($P < 0.0001$), period (pre-treatment (−60 to 0 min); post-treatment (0 to 60 min); $P < 0.0001$), and an interaction of treatment and period ($P < 0.0001$) on area under the curve (AUC) of plasma LH concentrations. AUCs with different superscripts differ ($p < 0.05$).

post-treatment period was greater than ($P < 0.05$) the AUC of LH in the pre-treatment period (−60 to 0 min) for FTM080:500 and FTM080:2500 (Fig. 2B).

## DISCUSSION

FTM080 was recently identified as a potent KISS1R agonist by structure–activity relationship studies on kisspeptin (*Tomita et al., 2007*; *Niida et al., 2006*). It has been reported that kisspeptin is inactivated by the cleavage of the Gly-Leu peptide bond in the C-terminal region by MMPs (*Takino et al., 2003*). Since kisspeptin and FTM080 share a common sequence (Phe-Gly-Leu-Arg) of the MMP-mediated cleavage site, FTM080 would be also deactivated by MMP-mediated digestion. However, the half-life of FTM080 in murine serum (6.6 h) is greater than that of KP-10 (completely digested within 1 h). Substitution of the Gly-Leu dipeptide moiety in FTM080 and KP-10 with appropriate dipeptide isosteres resulted in peptides (e.g., FTM145) resistant to degradation

by MMP-2 and -9, more stable in murine serum (e.g., compound FTM145 half-life = 38 h), while maintaining bioactivity for KISS1R *in vitro* (*Tomita et al., 2008*). However, *in vivo* activities of these peptides were not previously studied. Studies on pentapeptides derived from C-terminal kisspeptin fragments have been mainly focused on the design of analogs with superagonistic properties *in vitro*. Many previously developed pentapeptides, with apparent full agonistic activity at the KISS1R in cellular models, have not been evaluated in terms of gonadotropin secretion *in vivo*. When *in vivo* studies are conducted those observations do not always agree with findings from *in vitro* experimentation. For example, the attributes of FTM145 observed *in vitro* (resistance to MMP activity, greater stability in murine serum, and comparable bioactivity for KISS1R) were not recognized/observed in terms of gonadotropin secretion *in vivo* as FTM145 had no effect on plasma LH concentrations in anestrous ewes (B Whitlock, 2009, unpublished data).

Results of the present study revealed that intravenous FTM080 stimulated an increase of plasma LH concentrations in anestrous sheep. In Experiment 1 (pilot study), plasma LH concentrations increased approximately 7-fold between 0 and 45 min following intravenous treatment with FTM080 (from 0.14 to 0.97 ng/mL). The magnitude and duration of the LH-response following treatment with FTM080 in Experiment 1 was similar to previous observations in ovariectomized sheep given comparable doses of KP-10 (*Caraty et al., 2007*; *Whitlock et al., 2010*).

In Experiment 2 (a comparison of the effects of FTM080 and KP-10 on plasma LH concentrations in sheep) KP-10 stimulated the greatest magnitude and duration of an increase in LH secretion (0.64 to 5.55 ng/mL and 45 min, respectively). *Caraty et al. (2007)* reported that an intravenous bolus of KP-10 of approximately half the molar dose used in this experiment (500 pmol/kg) increased concentrations of LH in plasma of seasonally acyclic ewes from 0.2 ng/mL to 8.0 ng/mL which was similar to the response observed here. Although FTM080 did elicit a comparable LH-response (0.76 to 4.58 ng/mL), the dose necessary was 10-fold greater than the dose of KP-10 used (5000 vs. 500 pmol/kg, respectively). Moreover, although the *in vitro* half-life of FTM080 was greater than KP-10 (*Tomita et al., 2008*), the duration of the LH-response following an intravenous dose of FTM080 was less than the duration following a 10-fold lesser dose of KP-10 (30 min vs. 45 min, respectively). The shorter duration of the LH-response following treatment with FTM080 might simply be related to a difference in LH peak response (timing and possibly amplitude). It appears that LH-response slopes following treatment with FTM080 and KP-10 have a similar decay (are parallel).

*In vitro* screening and assays are useful to select agonist analogs for further *in vivo* studies. By improving biological stability while maintaining *in vitro* agonistic and receptor binding activity to KISS1R, *Asami et al. (2013)* identified the potent kisspeptin agonist analogue, TAK-683. TAK-683 was administered in several mammalian species including goats (*Tanaka et al., 2013*) and men (*Scott et al., 2013*) demonstrating excellent gonadotropin releasing activity *in vivo* at low doses. However, *in vitro* and *in vivo* activity/potency of KISS1R agonists do not always agree (*Gutierrez-Pascual et al., 2009*; *Curtis et al., 2010*). For instance, while KP-10 analog [dY][1]KP bound to the KISS1R with

a 4-fold lower affinity and had similar potency *in vitro* it had a more potent effect (4-fold) on LH than KP-10 *in vivo* (*Curtis et al., 2010*). Alternatively, another KP-10 analog, ANA5, bound with higher affinity to the KISS1R than kisspeptin but it was not more potent *in vitro* and less potent *in vivo* than KP-10 (*Curtis et al., 2010*). Thus although some kisspeptin analogs may act as KISS1R superagonists in specific *in vitro* systems, they may not have greater activity than kisspeptin *in vivo*.

It is interesting to speculate on mechanisms for the different responses obtained between FTM080 and KP-10 and why responses to FTM080 in sheep do not agree with those observed *in vitro*. There is the possibility that shorter kisspeptin analogs (FTM080 is a pentapeptide) have some limitation in terms of efficacy. The C-terminal amino acids of KP-10 (decapeptide that is a biologically active C-terminally amidated cleavage fragment of kisspeptin) are critical for efficient KISS1R binding (*Roseweir et al., 2009*) resulting overall in a greater focus on the screening of decapeptide instead of pentapeptide analogs of kisspeptin as potential KISS1R agonists. The difference in response observed here might also be the result of the animal model used for the *in vivo* experimentations. While previous *in vitro* assays to determine the bioactivity of FTM080 were conducted with human KISS1R (*Tomita et al., 2007*; *Niida et al., 2006*) the activity of FTM080 to sheep KISS1R has not been investigated. Contrarily, similar doses of kisspeptin have been administered to various species and various routes resulting very often in similar and comparable responses (*Seminara, 2005*). Differences in tissue distribution of KP-10 and FTM080 may be another possibility to explain differential *in vivo* efficacy. Only centrally, but not peripherally, administered KP-10 increased serum concentrations of growth hormone in sheep (*Whitlock et al., 2010*). Likewise, only centrally, but not peripherally, administered KP-10 induced c-Fos in GnRH neurons, suggesting that differential site of action of kisspeptin causes differential gonadotropin releasing efficacy *in vivo* (*D'Anglemont de Tassigny et al., 2008*). Differences in tissue distribution, especially at the hypothalamus, of FTM080 and KP-10 were not determined in this study. Pharmacokinetic profiles, including but not limited to clearance of FTM080, is another possible explanation for the different LH responses observed between FTM080 and KP-10. Higher clearance of FTM080 from the sheep circulation than KP-10 could be hypothesized to rationalize the lesser *in vivo* activity.

In conclusion, these data provide evidence to suggest that FTM080, a KISS1R agonist, stimulates the gonadotropic axis of ruminants *in vivo*. However, the increased half-life and comparable efficacy of FTM080 to KP-10 *in vitro* (*Tomita et al., 2008*) does not appear to translate to longer duration of efficacy *in vivo* in sheep.

### Funding

This project was supported by the Center of Excellence in Livestock Diseases and Human Health from the University of Tennessee, College of Veterinary Medicine. The funders had no role in study design, data collection and analysis, decision to publish, or preparation of the manuscript.

## Grant Disclosures

The following grant information was disclosed by the authors:
Center of Excellence in Livestock Diseases and Human Health from the University of Tennessee, College of Veterinary Medicine.

## Competing Interests

The authors declare there are no competing interests to report.

## Author Contributions

- Brian K. Whitlock conceived and designed the experiments, performed the experiments, analyzed the data, wrote the paper, prepared figures and/or tables, reviewed drafts of the paper.
- Joseph A. Daniel performed the experiments, analyzed the data, reviewed drafts of the paper.
- Lisa L. Amelse reviewed drafts of the paper.
- Valeria M. Tanco wrote the paper, reviewed drafts of the paper.
- Kelly A. Chameroy performed the experiments, wrote the paper, reviewed drafts of the paper.
- F. Neal Schrick contributed reagents/materials/analysis tools.

## Animal Ethics

The following information was supplied relating to ethical approvals (i.e., approving body and any reference numbers):
(1) Berry College Institutional Animal Care and Use Committee. (2) 2011/12-010.

## Data Availability

All raw data is available in Supplemental Information 1.

## Supplemental Information

Supplemental information for this article can be found online at http://dx.doi.org/10.7717/peerj.1382#supplemental-information.

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
