# Peer review of "Kisspeptin receptor agonist (FTM080) increased plasma concentrations of luteinizing hormone in anestrous ewes"

_PeerJ, doi:10.7717/peerj.1382_

## Round 0.1 · original submission · Major Revisions

· Academic Editor

Major Revisions

Please consider all the suggestions made by the reviewers in the revised version of your manuscript.

Reviewer 1 ·

Basic reporting

The article meets the basic reporting criteria even though the introduction would benefit from a more thorough analysis of the literature and the inclusion of some more details about the potential interest of the study.

Additional points that should be considered are the following:

It could be useful to mention at the end of the introduction which molecule has been used for the study.

In the introduction it is mentioned that Kp half-life is short, however references indicate to support this point are indirect or not pertinent (in the article of Hori et al. 2001 I was unable to find an indication about Kp half-life). A direct measure of Kp half-life in blood serum could be found in Chan et al., J Clin Endocrinol Metab, 2011, 96(6):E908-E915.

Page 10 of the discussion the FTM080 compound is misspelled TOM080.

The meaning of the third sentence of page 10 is unclear to me.

It is not clear to me what does it means: “…differences in tissue distribution of KP10 and FTM080 may be another possibility to explain differential in vivo efficacy”. Do the Authors refer to possible differences in blood brain barrier permeability of the 2 molecules as seems to be the case by the ensuing part of the discussion?

There is some approximation in certain phrases such as: “..FTM080 stimulated plasma LH concentration” I guess that FTM080 stimulated an increase of LH plasma concentration.

The graphs of figures are mixed with figure 1B coming after figure 2A.

Experimental design

Major comment. In the introduction is indicated that the molecule used has been modified to reduce degradation. Actually the major degradation site of the pentapeptide FTM080, the 7G8L amide bond, has not been modified as it is clearly indicated in reference 9. Authors acknowledge this fact in the discussion and indicate that another molecule with this modification has been synthesized, FTM145. Authors should explain why to perform their study they decided to use FTM080 that is not resistant to matrix metalloproteinase degradation instead that FTM145 that is resistant to matrix metalloproteinase degradation, has a longer half-life and good in vitro activity.

Minor comments. It would be important to precise the months of the year of the non-breeding season for this particular sheep breed. For some breeds June is the end of the non-breeding season and reproductive system could be starting to reactivate.

In experiment 1 the duration of the progesterone monitoring period is not indicated but only that samples were taken every 7 days.

Even though one can guess that test drug has been diluted in water it would be more appropriate to clearly indicate this point.

Some more details on the RIA method applied to measure LH would be useful.

For experiment 2 I think that reporting the means of LH plasma concentration over a 4-hour period after the treatment is confusing. It is quite clear that there is a strong albeit short lasting increase after treatment. Considering that the 4-hour period used is completely arbitrary if a longer period would have been used, for example 10 hours, probably no difference in the mean would have been observed despite a clear effect.

Validity of the findings

The results obtained are clear and convincing despite the limited number of animals used.

·

Basic reporting

In my opinion, this paper is well-written and conceptually easy to follow. The work is well done and the results are clear and fully support the conclusions. With regard to formatting, from what I understand, lines within the paper were not numbered and references were formatted as numbers within the text. Nonetheless, this was not an issue and the paper was very easy to read.

Experimental design

Design and analyses were appropriate. I would suggest moving the breed of sheep used to the second sentence of the Materials and Methods and removing it from the experimental descriptions. Also, it seems a little odd that the work seems to have been done at Berry College, but that the only one of the authors affiliated with that school is the second author. Is that correct? In addition, while it’s stated in the results, I think it would be useful to add the number of animals in each treatment group to the experimental descriptions in the Materials and Methods section. Finally, is there a way to compare the slopes of the decay following the kp-10 and 5000 pm injections of FTM080? The statement is made in the discussion that the duration of the LH response was less with FTM080 than kp-10, but that seems to be solely due to the differences in peak response achieved followed by similar decays. At the end of the discussion, higher clearance is listed as a possibility, but that doesn’t seem to be supported by the data herein when one compares the decays of the two responses (which seem parallel).

Validity of the findings

Findings were clear and seemed to be correctly interpreted and supportive of the conclusions.

Additional comments

In the abstract, it’s stated that FTM080:500 increased plasma LH compared to vehicle, but it’s not quite clear if that refers to mean LH concentrations or peak concentrations following injection. This is also not clear in the text of the results where it says mean plasma LH was 0.97 after treatment for analog-treated animals, but this seems to be the peak of the response and not the mean for the hour-long treatment period. Some clarification would be helpful. Also, I found it a little confusing that the introduction talked about alterations in these analogs that rendered them resistant to MMP inactivation, but then in the discussion it seems that FTM080 is not altered in that way (and thus maybe wouldn’t be expected to exhibit an extended half-life).

---

## Round 0.2 · Minor Revisions

· Academic Editor

Minor Revisions

Please, consider the 2nd reviewer's suggestions in the revised manuscript.

Reviewer 1 ·

Basic reporting

The article meets the basic reporting criteria and it reads well.

Experimental design

The experimental design meets the required standards

Validity of the findings

The results obtained are clear and convincing.

Additional comments

The authors have thoroughly taken in to account all the remarks and questions about the experimental design and all other suggestions that I felt would have improved their manuscript.

·

Basic reporting

No issues.

Experimental design

No issues

Validity of the findings

No issues

Additional comments

This is a resubmission. The authors have addressed many of my former concerns, but I would make the following suggestions based on the revised manuscript.
1. The first paragraph reads as if we now understand how all of this works. The discovery and investigation of kisspeptin has provided important insight, but there are still a lot of unknowns. I would suggest revising the first paragraph to reflect that.
2. Line 61 - just because it causes release doesn't necessarily infer an important physiological role. I would suggest adding the fact that antagonists reduce GnRH/LH secretion and use the Roseweir reference in that regard.
3. Line 79 - this is a bit confusing with the way it's written. Only one is modified, but both have extended half-life. Maybe using and/or instead of just and, then deleting the "respectively".
4. Line 141 (and elsewhere) - I realize that the authors were just trying to respond to a review comment, but I think they can just say "n=3 per treatment group) and delete the text as it is redundant.
5. Line 168 - this still reads as if this is for the entire treatment period, but the data refers only to the 15 minute time point. I would suggest specifically stating that LH secretion 15 minutes postinjection was......
6. Line 201 - suggest deleting "And" from the beginning of the sentence.
7. Line 215 - maybe reword to say "of an increase in LH secretion."

---

## Round 0.3 · accepted · Accept

· Academic Editor

Accept

Thank you for publishing in PeerJ.